# Electrodialysis for the Concentration of Lithium-Containing Brines—An Investigation on the Applicability

**DOI:** 10.3390/membranes12111142

**Published:** 2022-11-15

**Authors:** Frank Rögener, Lena Tetampel

**Affiliations:** 1Institute of Chemical Process Engineering and Plant Design, Technische Hochschule Köln, 50679 Köln, Germany; 2Linde GmbH Magyarországi Fióktelepe, 1097 Budapest, Hungary

**Keywords:** electrodialysis, lithium, brine recovery, membrane technology, lithium-ion batteries

## Abstract

The importance of lithium as a raw material is steadily increasing, especially in the growing markets of grid energy and e-mobility. Today, brines are the most important lithium sources. The rising lithium demand raises concerns over the expandability and the environmental impact of common mining techniques, which are mainly based on the evaporation of brine solutions (Salars) in arid and semiarid areas. In this case, much of the water contained in the brine is lost. Purification processes lead to further water losses of the ecosystems. This calls for new and improved processes for lithium production; one of them is electrodialysis (ED). Electrodialysis offers great potential in accessing lithium from brines in a more environmentally friendly way; furthermore, for the recovery of lithium from spent lithium-ion batteries (LIB), electrodialysis may become a vital technology. The following study focused on investigating the effect of varying brine compositions, different ED operation modes, and limiting factors on the use of ED for concentrating lithium-containing brine solutions. Synthetic lithium salt solutions (LiCl, LiOH) were concentrated using conventional ED in batch-wise operation. While the diluate solution was exchanged once a defined minimum concentration was reached, the concentrate solution was concentrated to the respective maximum. The experiments were conducted using a lab-scale ED-plant (BED1-3 from PCCell GmbH, Germany). The ion-exchange membranes used were PCSK and PCSA. The treated solutions varied in concentration and composition. Parameters such as current density, current efficiency, and energy requirements were evaluated. ED proved highly effective in the concentration of lithium salt solutions. Lithium chloride solutions were concentrated up to approximately 18-fold of the initial concentration. Current efficiencies and current densities depended on voltage, concentration, and the composition of the brine. Overall, the current efficiencies reached maximum values of around 70%. Furthermore, the experiments revealed a water transport of about 0.05 to 0.075% per gram of LiCl transferred from the diluate solution to the concentrate solution.

## 1. Introduction

Lithium demand is projected to triple by 2025 compared to 2018 [1], putting pressure on suppliers to increase their production. The predominant approach for lithium production from brine resources is an open pond evaporation process [2]. Concurrent with the rising lithium demand, concerns over the expandability and the environmental impact of this common mining technique are rising as well [3]. These trends call for new and improved processes for lithium production; one of them is electrodialysis (ED).

During the conventional evaporation process, which takes about 12 to 18 months, the brine is concentrated through solar evaporation [2]. During this time, the concentration of high lithium content brines rises from about 300 ppm to about 5000 ppm [3]. Other minerals contained in the brine, such as sodium, potassium, boron, and magnesium, are recovered or removed in the process as well. The composition of exemplary brine deposits, already used for commercial lithium extraction, is shown in Table 1.

Evaporation techniques are particularly of economic interest in countries with a dry and warm climate, high sun exposure, little rain, no strong winds, and low air humidity [2]. Due to the necessity for these climatic and regional conditions, only a few of the global brine resources are effectively used for commercial lithium extraction [2]. If evaporation is applicable, much of the water contained in the brine is lost [5]. Some alternative technologies aim at reinjecting the lithium-depleted brine back into the Salar, to decrease the environmental impact of lithium mining; however, this concept is still under investigation [5]. Novel processes have the potential to address these shortcomings. Accordingly, lithium reserves that are not yet exploitable could be assessed, while lowering the environmental impact. Currently, researchers, as well as companies, are exploring different technologies. These include ion-exchange processes [6], electrodialysis [7,8], or other membrane-based processes, such as nanofiltration [3,6,7] or membrane distillation, for combining water evaporation with membrane extraction for Li recovery [9].

ED offers great potential in the concentration of lithium-containing brine solutions. ED is a membrane process that allows for the concentration and depletion of ions in aqueous solutions using a direct electric current [10]. This way, an ion-enriched concentrate solution as well as an ion-depleted diluate solution are obtained. In the case of brine concentration, this could not only produce higher concentrated lithium solutions, but also water with low ion concentrations as a byproduct. Furthermore, ED is not dependent on climatic or territorial conditions. It can be operated using renewable energy sources, such as solar and wind power [11].

Few researchers have investigated the recovery of lithium from lithium-containing brines. For example, different membrane preparation techniques [12,13] or the combination of ED with other unit operations [14,15] have been applied. Membranes used for the ED process are either monovalent-ion selective membranes or bipolar membranes. Monovalent-ion selective membranes separate lithium-ions from multivalent ions, which are both found in naturally occurring brines (see Table 1). Examples of this type of membrane used for lithium recovery in lab-scale experiments are the Selemion CSO membrane or a membrane from ASTOM [16]. Another researcher successfully prepared membranes impregnated with ionic liquids for the selective recovery of lithium [12,17,18]. Bipolar membranes, on the other hand, are used for the production of lithium hydroxide from salt solutions [16,19,20].

Due to the high salinity and especially the calcium and magnesium content, the tendency for membrane fouling is high [9].

In the future, spent lithium-ion batteries (LIBs) will play a more important role as a secondary raw material for lithium extraction [21]. After grinding the spent LIBs, hydrometallurgical processes, including leaching with organic or inorganic acids, have been proposed for the separation and purification of Li [22]. ED was successfully applied for the recovery of lithium from spent lithium-ion battery (LIB) solutions in lab-scale experiments. However, only very few researchers have addressed the topic so far [8,23].

Accordingly, there is a demand for further investigations into the purification and concentration processes for lithium-containing solutions. The goal of this study is to show the general applicability of electrodialysis for the concentration of Li-containing solutions using cell pairs of PCSK (strongly acidic cation exchange membrane) and PCSA (strongly alkaline anion exchange membrane) membranes. In particular, the following parameters are important for the assessment: current density, current efficiency, maximum concentrations of lithium in the concentrate, energy consumption, and the influence of other components in the solution. This study will give further insights into how these parameters affect ED performance when using it for the separation of lithium-containing brines.

## 2. Materials and Methods

### 2.1. Test Plant and Set-Up

For the experiments, a lab-scale ED-plant BED1–3 from PCCell GmbH, Germany, was used. The plant consists of feed tanks for the concentrate, diluate, and electrode rinse solution of the ED stack; a DC power supply; and pumps, valves, and prefilters (Figure 1).

The membranes used in the ED stack are strongly acidic and strongly basic ion-exchange membranes from PCCell GmbH, Germany. The characteristics of the membrane stack are as follows:Effective membrane area (A_eff_) = 64 cm^2^;Spacer thickness (d_spacer_) = 0.45 mm;10 cell pairs, including 9 PcSK membranes, 10 PcSA membranes, and 2 PcMPE membranes (next to the electrodes);A plate-and-frame heat exchanger connected to a thermostat kept the solution temperatures at 21 ± 0.5 °C.

### 2.2. Measurement

Conductivity and temperature were measured using a JUMO CTI-500 conductivity/concentration and temperature transmitter, which is integrated in the test plant. Tolerances are ≤0.5% for the conductivity/concentration transmitter and ≤0.5% of the range for the temperature transmitter, respectively. Further conductivity measurements were done using a multiparameter meter type HI 9829, Hanna Instruments. Its accuracy is ±1% of reading or ± 1 µS/cm, depending on which value is greater. The scale used for weighing the salts was from Mettler, type PM4800 DeltaRange (Max 4100 g/800 g; d = 0.1 g/0.01 g). A volumetric flask, 5000 mL (±1.2 mL), was used for the preparation of the aqueous salt solutions.

### 2.3. Materials Used

Chemicals used for the experimental investigations are as follows:Lithium chloride (≥98.5%, pure), Carl Roth GmbH + Co. KG;Lithium hydroxide, as lithium hydroxide monohydrate (≥56.5% LiOH), Carl Roth GmbH + Co. KG;Sodium sulfate (≥99%, anhydrous), Carl Roth GmbH + Co. KG;Sodium chloride (>99.8%), Carl Roth GmbH + Co. KG;Potassium chloride (>99.5%), Carl Roth GmbH + Co. KG;Calcium chloride (anhydrous);Demineralized water (reverse osmosis-based water supply at TH Köln).

### 2.4. Operation of the Test Plant

The operation of the test plant was done in batches. The diluate solutions were exchanged once the conductivity reached a minimum value of 0.08 mS/cm. The concentrate solution was kept in the concentrate cycle throughout the respective investigations.

### 2.5. Correlation of Measured Values with Concentration

The conversion from conductivity to concentration for LiCl, NaCl, KCl, and CaCl_2_ was done using measured values gained by McCleskey [24]. The conversion of LiOH conductivity to molar concentration was based on values obtained by measurements done with an HI 9829 multiparameter meter.

### 2.6. Experimental Investigation Using Synthetic Solutions

The first set of experiments examined the performance of the electrodialysis on the concentration of different lithium salts. The advantage of using pure lithium salt solutions is the easy analysis of the test results and the absence of influencing ions.

Additionally, the transport behavior of other ions present in naturally occurring lithium-containing brines was investigated using synthetic solutions, too. This way, the influence of the anion as well as the cation on the concentration using ED could be analyzed.

Synthetic solutions comprised LiCl, LiOH, NaCl, KCl, and CaCl_2_ at concentrations varying between 1 g/L and 3 g/L. The applied voltages ranged between 10 and 20 V.

### 2.7. Quality Parameters

The following equations were used to describe the plant’s performance:

The electric charge, *C*, is calculated by Equation (1), as follows [25]:(1)$$C=\;\int_{0}^{\Delta t}I\left(t\right)dt=\bar{I}*\;\Delta t\;\left[As\right].\;$$
with *t* being the time and $\bar{I}$ being the average current (in A) during the time interval 0-*t* (in s).

Current efficiency (CE), *η*, is an important parameter to demonstrate which share of the applied current is transferred into ionic transport during the separation process. It is a percentage value, based on the actual passage of ions divided by the theoretical one, calculated through Faraday’s law (Equation (2)) [26]:(2)$$\eta =\;\frac{z*F*n}{N*\sum_{t=0}^{t}\left(I_{t}*\Delta t\right)}*100\%\;\left[\%\right]$$
where *z* is the valence of the ion; *n* is the number of transported ions (in moles); *F* is Faraday’s constant; *N* is the number of cell pairs; and *t* is the time in seconds.

*e_des_* is the specific electrical energy required for the transfer of ions from 1 L of diluate to the concentrate solution; it is calculated as follows [27]:(3)$$e_{des}=\frac{I_{St}*U_{St}*t}{\Delta n_{Dil}}\;\left[\mathrm{kWh}/\mathrm{mol}\right]$$
where *e_des_* stands for the specific desalination energy (in kWh/mol); i.e., the energy needed for the transfer of ions; *I_St_* is the current flowing through the stack (in A); *U_St_* is the voltage applied to the stack (in V); ∆*n_Dil_* is the transported L (in L); and *t* is the time (in s).

## 3. Results

### 3.1. Pure Lithium Solutions

To examine the maximum possible concentrate concentration, a long-term investigation was conducted for about 17.5 h with initial diluate and concentrate concentrations of 3 g/L LiCl. For security reasons, each run was stopped at the end of a working day and continued the next day.

Further experiments with LiCl and LiOH solutions were performed with four batches of the diluate solution. Each batch contained approximately 0.8 L of the respective aqueous solution.

#### 3.1.1. Long-Term Behavior of LiCl Solutions during Electrodialysis

After about 17.5 h, a concentrate concentration of 52.4 g/L LiCl was achieved (see Figure 2).

The estimated concentrations of LiCl equate to a lithium content of the initial brine of 0.4912 g/L Li and a final content of 8.5788 g/L Li.

Volume increase in the concentrate solution due to water transport caused by the hydrated ions was a limiting factor for the experiment. Especially, the hydrated form of the Li^+^ ion is relevant, as cations generally show higher hydration numbers [28]. Per 1 g/L LiCl transported through the membrane, the volume increase of the concentrate solution is about 0.02 L.

The current efficiencies during the long-term investigation with LiCl calculated for periods of 30 min are shown in Figure 3.

In general, a trend of decreasing current efficiency as a function of time (and concentrate concentration, accordingly) is visible in Figure 3, which can be attributed to several effects that cannot be quantified within the frame of the current investigations. These are especially: increasing coion transport through the ion exchange membranes, leading to decreasing permselectivity); ionic water transport; and increasing osmotic pressure of the concentrate solution, leading to back transport of water into the diluate. The depicted fluctuation of the current efficiency is attributed to the fact that some periods may include different numbers of batches. Furthermore, each time the run was stopped and continued only the next day, there was a slight drop in the concentrate concentration, which also influences the calculated CE. The concentrate drops are likely to be caused by osmotic back transport during downtime.

The mean value of the calculated CE shown in Figure 3 is 64.9%, the median is 64.2%, the standard deviation is 6.5, and the variance is 43.5.

Both the voltage and the concentration of the initial solution affect the current efficiency of the ED (Figure 4)**.** The initial concentration ranged between 1 g/L LiCl and 3 g/L LiCl. The voltages investigated ranged between 10 V and 20 V.

Figure 4 shows that a low voltage and low LiCl concentration in the solution lead to higher CEs. The change in concentration has a bigger effect on the CE than the change in the voltage. On average, changing the concentration from 1 g/L to 3 g/L resulted in a decline in CE of 7.0%, whereas the change in voltage from 10 V to 20 V led to a decline of 2.7%, on average.

#### 3.1.2. Comparison of the Behavior of Synthetic LiOH and LiCl Solutions

In further investigations, a lithium hydroxide solution was compared to solutions containing LiCl. In Figure 5, this comparison is shown for solutions containing 1 g/L of the respective solution.

The gradient of the LiCl curve is steeper, reaching higher concentrations at the same electrical charge compared to the investigations with LiOH. Accordingly, the transport of LiCl through the membrane appears to be better than of LiOH. This is also reflected by the calculated CE: the overall CE of the 1 g/L LiCl solution was 67.1% and of the 1 g/L LiOH solution 52.4%.

The current densities for LiCl ranged between 0.63 to 11.09 mA/cm^2^ and for LiOH between 0.94 to 16.88 mA/cm^2^.

Figure 6 shows the comparison between the 2 g/L LiOH and LiCl solutions.

Again, the slope of the solution with LiCl is steeper and the transport through the membrane seems better. The CEs support this trend, with the overall CE for the 2 g/L LiCl solution at 61.3% and the 2 g/L LiOH solution at 41.1%. While the concentration increase in LiCl is similar for both initial concentrations, LiOH shows a different behavior: the concentration increase in the more diluted solution (initial concentration 1000 ppm) is much stronger than in the more concentrated solution (initial concentration 2000 ppm). This indicates an influence of the pH on the membrane performance.

Especially OH^-^ ions differ significantly in their transport through AEMs compared to other anions, such as Cl^-^. Ionic conductivity of the hydroxide ion, when part of the AEM, depends largely on its level of hydration. When the water content is high, the ionic conductivity for the hydroxide is high as well; however, if the hydration is low, its conductivity drops lower than that of chloride [29]. This influence on ionic mobility may explain why the solution containing LiCl showed better results.

The current densities of the LiCl solution ranged between 0.63 to 14.84 mA/cm^2^, and for the LiOH solution between 1.25 to 20.47 mA/cm^2^.

#### 3.1.3. Energy Demand

Figure 7 shows the specific energy demand of the desalination of 1 L of LiCl and LiOH solutions at different initial concentrations. The data were calculated based on Equation (3).

It becomes obvious that the resulting specific energy demand for the LiCl and LiOH solutions are comparable. Extrapolating the curves, the energy needed for the desalination of one mol Li^+^ via ED at 20 V is between 214 kJ and 245 kJ (0.06 kWh/mol to 0.07 kWh/mol).

### 3.2. Comparison of the Behavior of LiCl, NaCl, KCl, and CaCl_2_ Solutions during Electrodialysis

Natural lithium-containing brine solutions generally contain impurities, as shown in Table 1. Accordingly, investigations of the behavior of some of the impurities during electrodialysis were conducted and compared to the behavior of LiCl. Figure 8 depicts the standardized concentration (c/c_0_) as a function of the electrical charge for aqueous LiCl, NaCl, KCl, and CaCl_2_ solutions during electrodialysis at the same process conditions.

The different cations influence the concentration behavior during electrodialysis. While the trends for KCl and CaCl_2_ are similar, the other curves are diverging from one another and LiCl shows the lowest slope of c/c_0_ as a function of the charge.

The current efficiencies were calculated through the measured concentrations (Table 2).

Water transfer through the membrane changed with the cation of the salt (Table 3). The solutions contained the same number of Cl- ions; thus, the solution containing a divalent cation (Ca^2+^) has a lower molar concentration of cations.

The water transfer depends on the hydration number of the ion, and divalent ions have higher hydration numbers than monovalent ions; thus, water transport of Ca^2+^ was higher [30]. In addition, Li^+^ has a higher hydration number than Na^+^ or K^+^. Accordingly, the volume increase in the Li-containing concentrate solution was the highest.

## 4. Discussion

The presented work attests to the suitability of ED as a good alternative to traditional treatment methods for the concentration of lithium-containing solutions. Traditional evaporation processes to concentrate lithium-containing brine are very low in their energy cost, as they mainly use naturally occurring and therefore free solar energy [31]. However, these processes can only be used for very few of the global lithium reserves due to natural conditions and they generally have a strong environmental impact. Thus, ED can potentially be used to treat lithium solutions, where traditional methods are not viable and for new applications, e.g., LIB recycling.

The effectiveness of electrodialysis with the present membranes (PCSK and PCSA) in terms of current efficiency was higher when the anions are chlorides instead of hydroxides. Furthermore, current efficiency was about 5% higher when the initial concentrations of the diluate and concentrate were low (1 g/L instead of 3 g/L).

With an initial concentration of 3 g/L LiCl (c_0_ = 0.491 g/L Li), the maximum concentration reached in the presented work was about 52 g/L LiCl (c_t_ = 8.58 g/L Li). This results in a concentration ratio (c_t_/c_0_) of about 1746%. Compared to values in the literature, this is a good value. Gmar et al. [32] report concentration ratios of about 396% in brine treatment and 3200% in the concentration of lithium hydrogen carbonate. Generally, concentrations >52 g/L LiCl could be achieved by electrodialysis, but the ionic water transport into the concentrate solution was a restriction in the present examinations.

The desalination energy demand per transferred mol of Li+ was about 0.06 kWh to 0.07 kWh. To put this number into context, energy consumption for separation of lithium-containing feed solutions found in the literature typically range from 0.0133 kWh/mol Li at the low end up to 3.8 kWh/mol Li for the different membranes tested [16]. Thus, the energy consumed during the present investigations is comparatively low. However, Nie et al. [33,34] achieved lower energy consumption using the Selemion CSO membrane.

Accordingly, the application of ED will mainly play a role in the concentration of brines for which traditional evaporation methods are not viable, such as Li recovery from spent LIBs.

Further investigations focused on the application of electrodialysis on synthetic solutions of typical impurities found in naturally occurring lithium-containing brines. It was shown that, especially for monovalent ions, the concentration characteristics were very similar for the different salts tested (LiCl, NaCl, KCl, and CaCl_2_). Particularly, the hydration numbers of the different dissolved cations seemed important for the process. They influence the transportation of water molecules attached to the respective cation through the membrane. On the contrary, when solutions with different anions were investigated, increased water transport through the membrane was not observed. Therefore, water transport could be a limiting factor when treating ions with high levels of hydration.

In conclusion, while this work has presented the general applicability of the PCSK and PCSA membranes for the electrodialysis of lithium-containing salt solutions, more research should focus on the utilization of industrial feed solutions, and the application of EDR technology to overcome the scaling-related problems with industrial solutions.

## Figures and Tables

**Figure 1 membranes-12-01142-f001:**
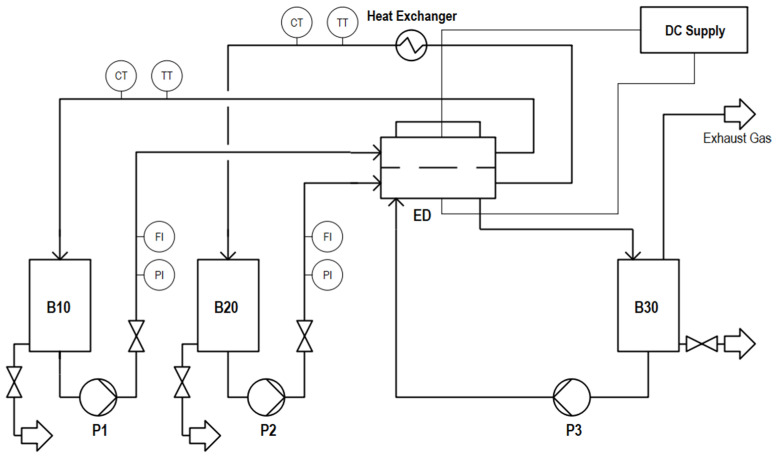
Flow chart of the ED lab plant.

**Figure 2 membranes-12-01142-f002:**
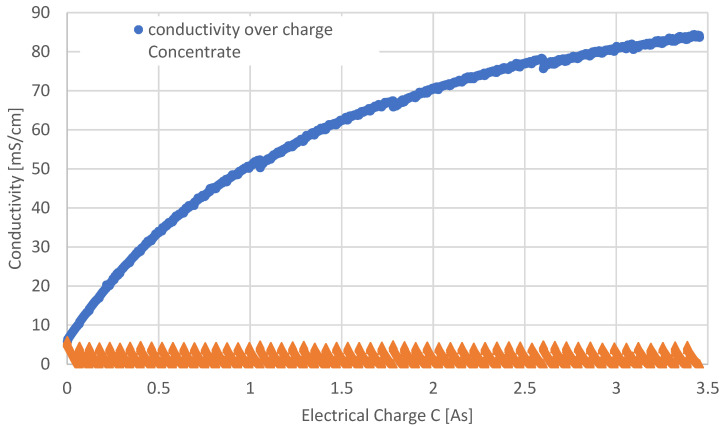
Conductivity during the long-term experiment over 17.5 h at ϑ = 21 °C as a function of the electrical charge. Initial diluate and concentrate concentrations were 3 g/L LiCl.

**Figure 3 membranes-12-01142-f003:**
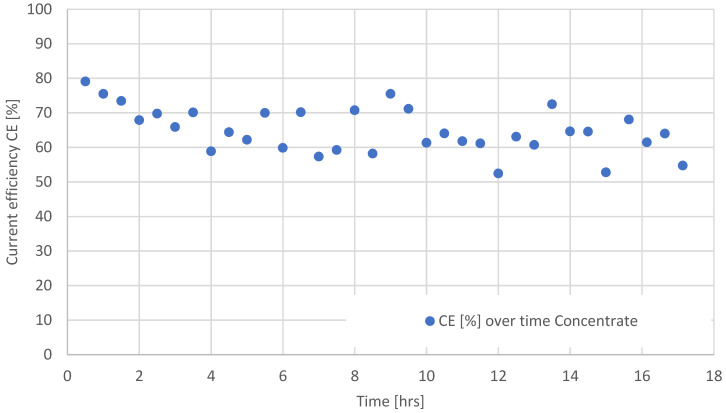
Current efficiency during electrodialysis at ϑ = 21 °C as function of time. Initial diluate and concentrate concentrations were 3 g/L LiCl.

**Figure 4 membranes-12-01142-f004:**
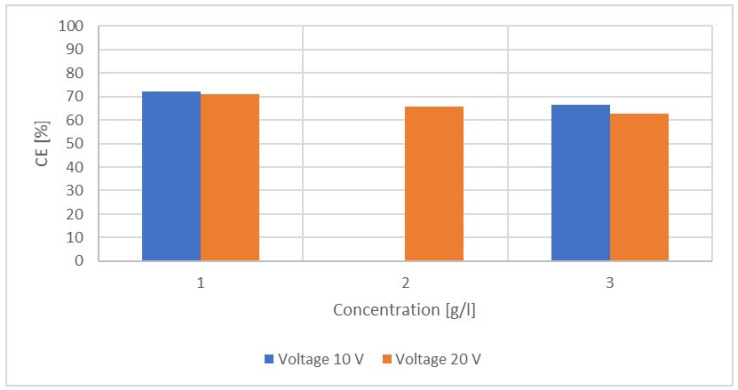
CE as a function of the concentration of LiCl in an aqueous solution for varying voltages at ϑ = 21 °C.

**Figure 5 membranes-12-01142-f005:**
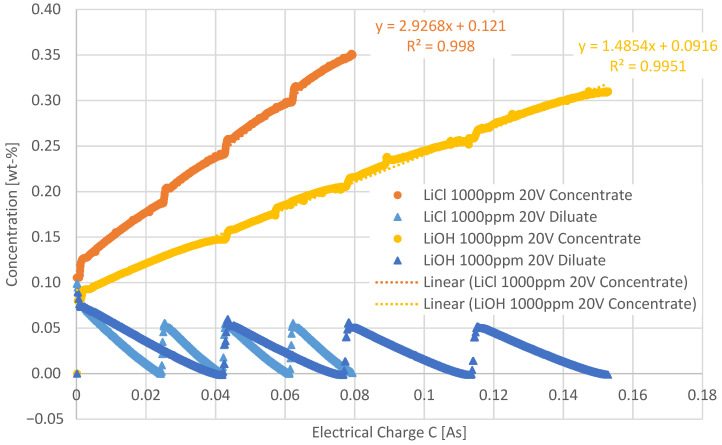
Comparison of the LiCl and LiOH concentrations during electrolysis at ϑ = 21 °C and 20 V as a function of the electrical charge (C). The initial diluate and concentrate concentrations were 1 g/L, respectively.

**Figure 6 membranes-12-01142-f006:**
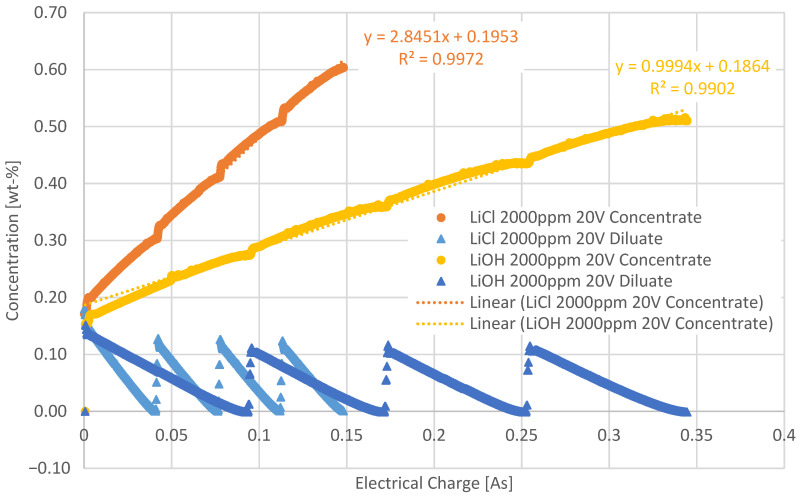
Comparison of experiments using initial concentrations of 2 g/L of the LiCl and LiOH solutions, respectively, at 20 V, as a function of the electrical charge at ϑ = 21 °C.

**Figure 7 membranes-12-01142-f007:**
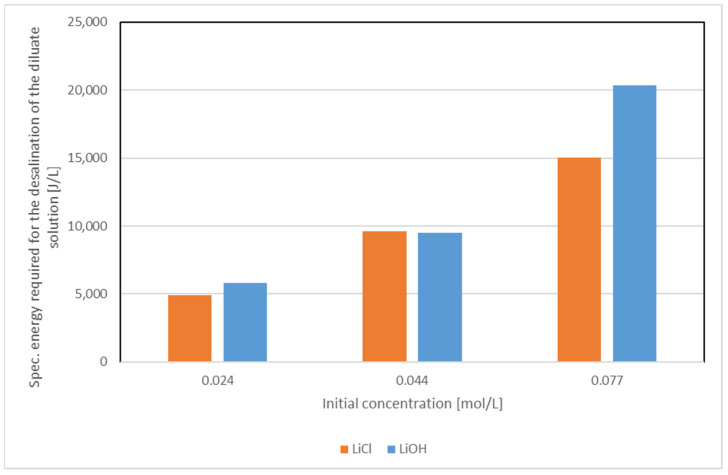
Comparison of the specific desalination energy required for varying molar concentrations of LiCl and LiOH solutions at ϑ = 21 °C.

**Figure 8 membranes-12-01142-f008:**
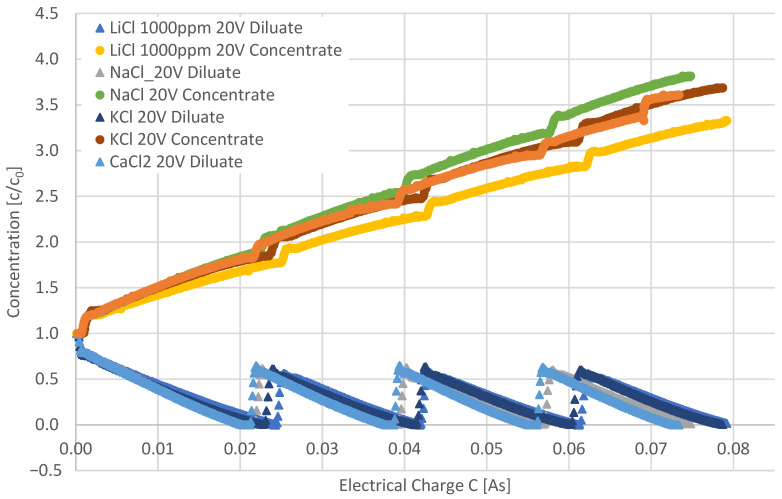
Comparison of investigations with LiCl, NaCl, KCl, and CaCl_2_ as a function of the electrical charge with adjusted values (c/c0) at ϑ = 21 °C.

**Table 1 membranes-12-01142-t001:** Selected typical lithium-bearing brine compositions; other elements and compounds may also be present [4].

Element	Clayton Valley, USA	Salar de Atacama, Chile	Salar de Hombre Muerto, Argentina	Salar de Rincon, Argentina	Zhabuye Salt Lake, China	Qaidam Basin Salt Lakes, China
All numbers in wt%						
Li	0.02–0.04	0.11–0.31	0.05–0.06	0.03	0.05–0.10	0.01–0.03
K	0.53–1.00	1.80–2.97	0.52–0.62	0.62–0.66	2.64–3.83	0.60–0.66
Mg	0.03–0.06	0.82–1.53	0.05–0.09	0.28–0.30	0–0.001	0.47–3.51
Ca	0.02–0.05	0.02–0.04	0.05–0.09	0.04–0.06	0–0.01	0.02–0.42
B	0–0.01	0.06–0.07	0.02–0.04	0.04	0.29–1.46	0.03–0.05
Na	6.20–7.50	1.03–9.10	9.79–10.30	9.46–9.79	10.66–10.81	6.20–6.84
Cl	10.10–11.70	2.03–18.95	15.80–16.80	15.80	12.16–12.31	9.20–20.42

**Table 2 membranes-12-01142-t002:** Comparison of the current efficiencies (CE), overall and per batch, calculated for the investigations with LiCl, NaCl, KCl, and CaCl_2_.

	LiCl Solution		NaCl Solution		KCl Solution		CaCl_2_ Solution	
	Diluate	Concentrate	Diluate	Concentrate	Diluate	Concentrate	Diluate	Concentrate
CE 1st batch	89.9%	71.8%	73.0%	69.5%	76.5%	60.5%	89.7%	65.5%
CE 2nd batch	61.3%	49.3%	57.5%	60.8%	59.7%	51.9%	70.8%	49.3%
CE 3rd batch	58.9%	50.4%	57.4%	56.5%	60.8%	45.4%	70.1%	45.6%
CE 4th batch	56.1%	46.6%	53.2%	55.1%	56.1%	49.0%	66.5%	58.2%
Overall CE	**-**	67.2%	**-**	64.3%	**-**	55.4%	**-**	58.7%

**Table 3 membranes-12-01142-t003:** Comparison of water transport through the membranes depending on the cation; investigations conducted at 20 V for concentrations for Cl^-^ ions of 0.024 mol/L at ϑ = 21 °C (cation molar concentration adjusted accordingly).

Feed	Volume of Diluate Solution Treated (L)	Volume Increase in Concentrate Solution (L)	Hydration Number of the Cation According to [24]
LiCl solution	3.3	0.025	5
NaCl solution	3.2	0.010	4
KCl solution	3.2	0.010	3
CaCl_2_ solution	3.2	0.015	6

## Data Availability

Data are contained within the article.

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
