# Peer review of "Electrodialysis for the Concentration of Lithium-Containing Brines—An Investigation on the Applicability"

_membranes, 2022, doi:10.3390/membranes12111142_

Round 1
Reviewer 1 Report
The study focus on using ED to concentrate Li. The result is clear and significant. My minor comments:
- Figure 3: linear line is not appropriate. The authors should replace with non-linear curve
- The results for LiOH (eg. Fig6) clearly highlight the issue of pH on the performance. The authors should comment on that for future development.
Author Response
Dear reviewer,
thank you for your valuable remarks. In the following you can find our responses to your comments:
1. Figure 3: linear line is not appropriate. The authors should replace with non-linear curve
In general, a trend of decreasing current efficiency as a function of time (and concentrate concentration, accordingly) is visible in figure 3, which can be attributed to several effects that, however, cannot be quantified within the frame of the current investigations. These are especially:
- increasing coion transport throught the ion exchange membranes that leads to decreasing permselectivity
- Ionic water transport (as described in our paper),
- increasing osmotic pressure of the concentrate solution that leads to back transport of water into the diluate.
This additional information was added in line 196ff.
Accordingly, an interpolation to determine a fit function was waived, as any fit function would only reflect a mathematical manipulation instead of referring to the occurring physical-chemical effects.
The results for LiOH (eg. Fig. 6) clearly highlight the issue of pH on the performance. The authors should comment on that for future development.
Line 242ff was added to explain more clearly the discovered pH effect: "While the concentration increase as a function of the charge is similar for both initial LiCl concentrations, LiOH containing solutions show a different behavior: The concentration increase in the more diluted solution (initial concentration 1000 ppm) is much stronger than in the more concentrated solution (initial concentration 2000 ppm). This indicates an influence of the pH on the membrane performance."
Reviewer 2 Report
In this manuscript “Electrodialysis for the concentration of lithium-containing brines-An investigation on the applicability”. In this paper, by increasing the research on electrodialysis, lithium can be obtained from brine in a more environmentally friendly way. This manuscript is meaningful piece for scientific advance. However, there are still minor issues with this manuscript, which should be addressed before it can be considered for publications. For example:
1. Please check the references and keep the formatting consistent.
2. Please add a paragraph in the introduction that describes the innovations of this paper.
3. Please organize the information in Figure 2 and Figure 4, we think it is necessary to draw a histogram to express their changes directly.
4. Please explain how to get Figure 7 and elaborate on the results
5. Some related references are recommended to be considered to cite:
Chem. Eng. J., 2020, 401, 126005.
Science China Chemistry, 2020, 63, 475-482
J. Colloid. Interf. Sci., 2020, 563, 328-335.
Author Response
Dear reviewer,
thank you for your valuable remarks. In the following you can find our responses to your comments:
1. Please check the references and keep the formatting consistent.
6 corrections of the reference list were carried out
2. Please add a paragraph in the introduction that describes the innovations of this paper.
In line 20, a passage describing special features of this paper, was added
3. Please organize the information in Figure 2 and Figure 4, we think it is necessary to draw a histogram to express their changes directly.
Figure 2:
To the authors, the design of figure 2 as an x,y diagram seems to be adequate, as it depicts the nonlinear course of the concentrate concentration as a function of the electric charge, which is due to undesired side effects, such as water transport and decreasing permselectivity. A transfer into a histogram would lose this information.
Figure 4
A change of the design of figure 4 is reasonable, as the transfer into a histogram does not affect the information included. Furthermore, the histogram contributes to a better understanding of the findings. Accordingly, the design of figure 4 was changed.
4. Please explain how to get Figure 7 and elaborate on the results
Equation 3 was adapted and a link to the equation was given in line 260
5. Some related references are recommended to be considered to cite
We checked the mentioned sources, see titles of the articles. We could not see any significant connections between the suggested papers - which are focusing on improvements of Zn-air batteries - and the topic of our article, which is based on the concentration of Li using electrodialysis.
- Chem. Eng. J., 2020, 401, 126005: Synthesis of confining cobalt nanoparticles within SiOx/nitrogen-doped carbon framework derived from sustainable bamboo leaves as oxygen electrocatalysts for rechargeable Zn-air batteries.
- Science China Chemistry, 2020, 63, 475-482: Porous phosphorus-rich CoP3/CoSnO2 hybrid nanocubes for high-performance Zn-air batteries
- J. Colloid. Interf. Sci., 2020, 563, 328-335: Ultrathin cobalt pyrophosphate nanosheets with different thicknesses for Zn-air batterie
Accordingly, the suggested articles were not taken into account for the reference list of the present paper.
Reviewer 3 Report
Review comments
This study investigated the Electrodialysis for the concentration of lithium-containing brines – An investigation on the applicability. This manuscript needs careful modification. I feel that this work could only be possibly recommended for publication after minor revision, improvement.
Some comments are shown as following:
1. The abstract can be improved. Also, please include background statement into the abstract.
2. What is the novelty of the study in comparison with the reference given? State the novelty of this work clearly at the end of introduction.
3. Why the authors not compared the obtained results to other similar published studies?
4. The results part can be improved by more discussion of the obtained results.
5. The authors should demonstrate that this is not an isolated case study. What are the general implications of the study? Comments on the general applicability and transferability of the results are needed.

Author Response
Dear reviewer,
thank you for your valuable comments.
1. The abstract can be improved. Also, please include background statement into the abstract.
Lines 14ff were added to stress the ecological background of our activities.
Lines 20ff were added to stress the procedure of the investigations.
2. What is the novelty of the study in comparison with the reference given? State the novelty of this work clearly at the end of introduction.
The novelties of this study include application of the membrane cells of PCSK and PCSA for the concentration of Li solutions. This was added in lines 93ff.
3. Why the authors not compared the obtained results to other similar published studies?
The authors added a comparison of the concentration ratio and spec. energy demands obtained in the present investigations with previously published findings (lines 309ff and 317ff). Accordingly, references 32-34 were added.
4. The results part can be improved by more discussion of the obtained results.
The findings of the investigations were explained more in detail (s. lines 187f; lines 199ff; 245ff). Also, chapter 4 - discussion - was extended.
5. The authors should demonstrate that this is not an isolated case study. What are the general implications of the study? Comments on the general applicability and transferability of the results are needed.
Chapter 4 - discussion - was extended. Special attention was given to further applications. The performance of membranes PCSA and PCSK proved to be comparable to previously investigated Selemion and Neosepta membranes.